# Infection Risk in Patients with Dermatomyositis Associated with Anti-MDA5 Antibodies: A Historical Cohort Study

**DOI:** 10.3390/biomedicines10123176

**Published:** 2022-12-08

**Authors:** Anne-Claire Billet, Thomas Barba, Frédéric Coutant, Nicole Fabien, Laurent Perard, Pascal Sève, Jean-Christophe Lega, Cécile-Audrey Durel, Laure Gallay, Arnaud Hot

**Affiliations:** 1Department of Internal Medicine, Édouard Herriot Hospital, Hospices Civils de Lyon, 5 Place d’Arsonval, 69003 Lyon, France; 2Immunology Department, Lyon South Hospital, Hospices Civils de Lyon, 165 Chemin du Grand Revoyet, 69310 Pierre Bénite, France; 3Immunogenomics and Inflammation Research Team, University of Lyon, Edouard Herriot Hospital, 5 Place d’Arsonval, 69003 Lyon, France; 4Department of Internal Medicine, Saint Joseph Saint Luc Hospital, 20 Quai Claude Bernard, 69007 Lyon, France; 5Department of Internal Medicine, La Croix Rousse Hospital, Hospices Civils de Lyon, 103 Grande Rue de la Croix Rousse, 69004 Lyon, France; 6Department of Internal and Vascular Medicine, Lyon South Hospital, Hospices Civils de Lyon, 165 Chemin du Grand Revoyet, 69310 Pierre Bénite, France; 7Evaluation and Modeling of Therapeutic Effects Team, UMR CRNS 5558, Claude Bernard University Lyon 1, 43 boulevard du 11 Novembre 2018, 69100 Villeurbanne, France

**Keywords:** dermatomyositis with anti-MDA5 autoantibodies, infectious complications, prophylaxis

## Abstract

Objective: Dermatomyositis associated with anti-MDA5 autoantibodies (DM-MDA5+) is a rare autoimmune disease usually characterized by skin involvement, often-severe lung involvement, and general features. Several reports of infections have been described, sometimes early after the introduction of immunosuppressive therapy. We studied the infection risk in a DM-MDA5+ population. Methods: A retrospective cohort study comparing the number and type of infections during the follow-up of 19 patients with DM-MDA5+ and 37 patients with another type of inflammatory myopathy was analyzed. Patients in both groups were matched on initial immunosuppressive therapy. We described and compared significant infectious complications (SIC) in each group. Results: Patients DM-MDA5+ had more SIC: 27 events in the DM-MDA5+ group versus 6 in the controls (HR 7.08, 95% CI 2.50–20.04, *p* < 0.0001). The number of SIC per patient was higher in DM-MDA5+ (1.4 ± 1.57 vs. 0.16 ± 0.44, *p* < 0.001). These were mainly lung (n = 13, 48%) and skin infections (n = 6, 22%), more often infections of an undetermined infectious agent (n = 11, 41%) or of bacterial origin (n = 9, 33%). A few cases of opportunistic infections were reported. The median duration of follow-up without SIC event in the DM-MDA5+ cohort was 3.5 months. Conclusion: Patients with DM-MDA5+ have an increased infection risk compared to others inflammatory myopathies irrespective of immunosuppressive therapy exposure. These results highlight the importance of monitoring for infection during patient follow-up.

## 1. Introduction

Long considered a single autoimmune disease with various presentations, inflammatory myopathy (IM) is now divided into multiple entities with clinical and biological specificities. Advances in immunology, corresponding mainly to identification of specific anatomopathological patterns and myositis-specific autoantibodies (Abs), have greatly helped the classification of these diseases.

The anti–melanoma differentiation-associated gene 5 (MDA5) Abs associated dermatomyositis (DM-MDA5+) is a rare disease characterized by skin ulcerations, polyarthralgia, severe interstitial lung disease, and moderate or even no muscle involvement. DM-MDA5+ patients have poor survival (67% at five years versus 92% for others IM), especially linked to rapidly-progressive interstitial lung disease [1,2,3]. 

Furthermore, anti-MDA5 Abs target the MDA5 protein, a key player in the innate immunity against viral and fungal infections. MDA5 is a cytosolic sensor that detects viral or fungal dsRNA and contributes to the response through the production of type I interferon (IFN-I) [4,5]. The role of Abs to the disease is currently unknown, as well as the impact of anti-MDA5 Abs on the defense against infectious agents [6].

Cohort studies showed an increased risk of opportunistic infections in patients with IM [7]. However, data is lacking regarding the DM-MDA5+ subgroup and whether patients with DM-MDA5+ present an increased risk of opportunistic infections has not yet been addressed. It can however be speculated that anti-MDA5 Abs could affect the establishment of an adequate immune response against infectious agents due to the central role played by MDA5 in the immune system. 

To investigate the risk of infections associated with DM-MDA5+ as compared with other subtypes of IM, a retrospective cohort study was conducted in a tertiary hospital in France.

## 2. Methods

### 2.1. Patients

This study is a retrospective cohort including adult patients diagnosed with DM between January 2000 and May 2020 in a tertiary hospital of Lyon (France). Inclusion criteria were age over 18 years old, DM diagnosis according to the Bohan and Peter [8,9] or American College of Rheumatology criteria [10], and anti-MDA5+ Abs. The detection of anti-MDA5 Abs was performed by using a specific line-immunoassay (Euroimmun, Lübeck, Germany) or a specific dot immunoassay (D-Tek, Mons, Belgium) according to manufacturer’s instructions [11]. This group was considered DM-MDA5+. A comparison group was set up composed of prevalent cases of different auto immune myositis (dermatomyositis, anti-synthetase syndrome, connective tissue-associated myositis, or immune-mediated necrotizing myopathy) grouped under the term IM-MDA5−. They fulfilled the same inclusion criteria apart from that of anti-MDA5 autoantibodies with a one-to-two ratio. Patients of both groups were matched on (i) age at diagnosis, (ii) gender, and (iii) first line therapy (Appendix A). Patient records and information were anonymized and deidentified prior to analysis. The study was approved by the institutional review board and the ethics committee of the Hospices Civils de Lyon.

### 2.2. Clinical Assessment

Medical charts were retrospectively reviewed to collect clinical and laboratory data, including demographic data, disease presentation, underlying immunosuppression factors, comorbidities, treatments, and prophylactic measures. 

Significant infectious complication (SIC) was identified by clinical manifestations, imaging findings, and positive microbiological tests from blood, and sputum, bronchoalveolar lavage (BAL) fluid culture [12]. SIC required the initiation of anti-infective treatment and/or hospitalization during the follow-up. Severe SIC was defined as an infectious event requiring intensive care unit (ICU) management or resulting in patient death. SIC of an undetermined infectious agent corresponded to events requiring the initiation of anti-infective treatment without microbiological documentation.

### 2.3. Statistical Analyses

Categorical variables were expressed as percentages and compared with exact Fisher’s or chi-squared tests. Continuous variables were expressed as mean ± standard deviation (SD) and compared using the Student’s test (*t*-test). Clinical and biological data of the two groups were compared using univariate models. 

Event-free survival was calculated from the date of DM or IM diagnosis until the first occurring SIC. Survival curves were built with the Kaplan–Meier method and compared with the log-rank test. The Cox proportional hazards regression model was used in univariate model to estimate hazard ratio (HR) and its 95% confidence interval (95% CI). All the tests were two-sided, and *p*-values strictly inferior to 0.05 were considered significant. Analyses were carried out using R software version 4.0.0 (R Foundation for Statistical Computing, Vienna, Austria, 2020, https://www.R-project.org/, accessed on 1 January 2020).

## 3. Results

### 3.1. Cohort Description

#### 3.1.1. Clinical Characteristics

Nineteen DM-MDA5+ patients and 37 IM with other Abs were included. As shown in Table 1, both groups were comparable at baseline regarding demographics and general comorbidities. The one-to-two ratio was respected for all cases except for one due to the absence of patient matching for the treatment (corticoids and immunoglobulins).

In the control group, there were 30 patients with anti-synthetase syndrome or connective tissue-associated myositis, six patients with DM, and one patient with immune-mediated necrotizing myopathy.

DM-MDA5+ patients had more weight loss (n = 9, 47% vs. n = 7, 19%, *p* = 0.05), more skin involvement with facial erythema (n = 8, 44% vs. n = 6, 17%, *p* = 0.06) and digital ulcerations in eight cases, and similar muscle involvement (n = 6, 32% vs. n = 12, 32%, *p* = 1) than the controls (Table 1). Interstitial lung diseases were observed in the same proportions between the two groups (DM-MDA5+ n = 11, 58% vs. IM-MDA5− n = 20, 54%, *p* = 1) with ground glass opacities (n = 5, 26% vs. n = 13, 36%, *p* = 0.66) or lung fibrosis (n = 6, 32% vs. n = 11, 20%, *p* = 1). The assessment of the respiratory function (defined by restrictive disorders or abnormalities of diffusing capacity for nitrogen monoxide) showed no difference between the two groups.

#### 3.1.2. Biological Characteristics

At baseline, the lymphocyte count (DM-MDA5+ 1106 ± 749.1/mm^3^ vs. IM-MDA5− 1472 ± 647.2/mm^3^, *p* = 0.11) was similar in both groups. There was no significant difference in immunoglobulin levels (DM-MDA5+ 13.4 ± 2.9 g/L vs. IM-MDA5− 12.2 ± 0.9 g/L, *p* = 0.30). However, neutrophil count was higher in the IM-MDA5− group (6607 ± 3234.6/mm^3^ vs. 4055 ± 3232.9/mm^3^, *p* = 0.02) (Table 1). 

#### 3.1.3. Cancer Associated

Percentages of concomitant cancer were similar in the two groups (DM-MDA5+ n = 4, 21% vs. IM-MDA5− n = 5, 14%, *p* = 0.73), as were the proportions of patients having undergone chemotherapy and radiotherapy (Table 1). The DM was considered as paraneoplastic in four cases. The two concomitant cancers in the test group were of lung and ovarian origin.

#### 3.1.4. Antibiotic Preventive Therapy

DM-MDA5+ patients received more pneumocystis prophylaxis than IM-MDA5− patients (n = 12, 63% vs. n = 8, 23%, respectively, *p* = 0.01), particularly by cotrimoxazole (n = 11, 58% vs. n = 7, 20%, *p* = 0.01). However, no difference regarding anti-pneumococcal vaccination (DM-MDA5+: n = 9, 47% vs. IM-MDA5− n = 12; 38%, *p* = 0.69) (Table 1) was observed.

#### 3.1.5. Immunosuppressive Therapy

As expected with the matching procedure, immunosuppressive regimens were comparable between the two groups (Appendix A). The three main therapeutic options used by physicians were corticosteroids alone (n = 6, 32% vs. n = 12; 33%), corticosteroids in combination with DMARDS (n = 5, 26% vs. n = 10; 27%), and corticosteroids with immunoglobulins (IVIG) (n = 2, 11% vs. n = 4; 11%).

Patients with DM-MDA5+ received more corticosteroid pulses (n = 6, 32% vs. n = 2; 5%, *p* = 0.02), although there was no significant difference for induction (0.95 mg/kg/d ± 0.12 vs. 0.95 mg/kg/d ± 0.25, *p* = 0.94) and maintenance of corticosteroid doses (9.97 mg/d ± 15.9 vs. 5.85 mg/d ± 7.7, *p* = 0.33).

### 3.2. Infections

#### 3.2.1. Prevalence of Infections

The mean follow-up time was 70 months. Thirty-three SIC occurred in 18 patients during follow-up (Table 2).

Infectious complications in the DM-MDA5+ group were of bacterial (*S. aureus*, *S. pneumoniae*, enterobacteria; n = 9 infections; 33%), viral (cytomegalovirus and varicella-zoster virus; n = 3.9%), and fungal (*P. jirovecii*, *Candida* sp., *A. fumigatus*; n = 4; 12%) origins or related to an undetermined infectious agent in 41% of the cases (n = 11). 

The sites of the infections were the lungs (n = 13, 48%), the skin (n = 6, 22%), bacteriemia (n = 3, 11%), the joints (n = 2.7%), the ENT sphere (n = 2.7%), or the urinary tract (n = 1, 4%). Eight (30%) infectious complications required ICU admission and led to the death of the patients in three cases (9%).

SIC had the same locations and microbiological sites in both groups except infections of an undetermined infectious agent that were more frequent in DM-MDA5+ patients. SIC of undetermined infectious agent were in majority lung infections (n = 7), occurring within 24 months of diagnosis of DM. These were characterized by clinical signs of pneumonia (cough, fever, congestion), a biological inflammatory syndrome, and radiological signs. The evolution was favorable after antibiotic therapy. Other SIC of an undetermined infectious agent were ENT or skin infections.

#### 3.2.2. Factors Associated with Infections

Thirteen patients (68.5%) in the DM-MDA5+ group presented SIC compared to five patients (13.5%) in the control group (*p* < 0.001). The SIC rate per patient in the DM-MDA5+ was greater than in the IM-MDA5− group (1.4 ± 1.57 vs. 0.16 ± 0.44, *p* < 0.001). Belonging to the DM-MDA5+ group was significantly associated with an increased risk of SIC (HR, 7.08, *p* < 0.001, Figure 1). The median SIC free time in the DM-MDA5+ group was 3.5 months.

Univariate analyses also identified history of cancer (HR 3.97, *p* = 0.01), maintenance corticosteroid dosage (HR, 1.06, *p* = 0.01), and male gender (HR 2.80, *p* = 0.04) as associated with increased risk of SIC, whereas evidence of ground glass lesions on chest CT appeared as a protective factor (HR 0.24, *p* = 0.02) (Appendix A). Neither *Pneumocystis jirovecii* prophylaxis nor anti-pneumococcal vaccination were associated with the risk of SIC occurrence.

## 4. Discussion

To the best of our knowledge, this study is one of the first to assess the infection risk especially of a DM-MDA5+ group. Our cohort has a large number of patients for a rare pathology in the Caucasian population [6]. The major strength of this work was an exhaustive collection of infectious events that occurred during the follow-up of DM-MDA5+ patients and a strict matching with other inflammatory myopathies on immunosuppressive first line therapy.

Our group of DM-MDA5+ patients had the same characteristics as those described in the literature with a predominance of extra muscular manifestations, such as weight loss (47%), skin lesions (90%), and lung damages (58%) [3]. Lung involvement appeared in the same proportions in two groups because of a majority of anti-synthetase syndrome in control group.

In a previous study, Marie et al. reported 33% of infections during follow-up of patients with myositis, including 12% of opportunistic infections that occurred mainly in the first year with high doses of corticosteroids [7]. Others authors describe infections that seem to occur more specifically in DM-MDA5+, such as pneumocystis and CMV infections [13,14,15], more currently observed with combined immunosuppressive therapy (cyclophosphamide and tacrolimus) [14]. Recently, a large cohort study demonstrated the prevalence of infections in patients with IM was 27.6%, and anti-MDA5 abs was identified as a contributing factor in the development of infections in patients with IM [16].

Our study also demonstrated DM-MDA5+ patients present an over risk of SIC during follow up compared to other forms of myositis, regardless of the immunosuppressive treatment. In addition, it describes the characteristics of these infections. SIC were most often infections of an undermined infectious agent or of bacterial origin and occurred early in the follow-up. Patients with ILD often developed lung infections, and digital ulcers were complicated by skin infections. These infections could be severe and required intensive care. However, we stopped the collection at the beginning of the COVID-19 epidemic, and we have no cases of COVID-19 infections.

We could not define the impact of pneumocystis prophylaxis in the occurrence of SIC because of a limited sample size. In addition, in our group, pneumococcal vaccination coverage was low. Nevertheless, in daily clinical practice, it seems important to propose prophylactic measures as prophylaxis against pneumocystis, influenza, and pneumococcal vaccination. 

Moreover, we could not assess the impact of the response to immunosuppressive therapy due to missing data from retrospective collection.

The pathophysiological mechanisms that may explain this over infection risk remains unknown. In DM-MDA5+, autoreactive B cell repertoire shows a bias towards the production of auto Abs that stimulate the secretion of IFN-gamma [11]. DM-MDA5+ is a Th1-driven disease, and the dysregulated production of IFN-gamma is linked to the severity of the disease [11]. A systemic dysregulation of T cell functions by auto Abs could therefore be the cause of severe tissue damage and could also interfere with the establishment of an efficient anti-infectious immune response.

The predisposition to infections of patients with DM-MDA5+ could also be explained by the role of MDA5, a key player which induces expression of IFN-I during the early stage of viral or fungal infection [5,17]. An inherited deficiency of the MDA5 protein promotes the development of severe respiratory infections (mostly viral but also bacterial) in children during the neonatal period [18,19,20]. Abs that target MDA5 and alter its function could then result in an inappropriate antiviral or antifungal response [5].

An alternative hypothesis that could potentially explain that DM-MDA5+ patients are more susceptible to infections could be the neutropenia observed in these patients. This can be explained in particular by the dysregulation of neutrophils that takes place in the disease. Indeed, it has recently been shown neutrophils from DM-MDA5+ patients are more prone to NETosis, a form of regulated cell death during which neutrophil extracellular traps (NET) and enzymes are extruded from the cell [21]. This abnormal NETosis appears to be induced by autoAbs, although the precise pathophysiological mechanisms are still poorly understood [21]. A dysregulation of this important process of the body’s defense against infectious agents could thus contribute to facilitating infections in DM-MDA5+ patients.

The current study had several limitations. First, the sample size was relatively small, which is explained by the fact that DM-MDA5+ is a rare disease, especially in Western countries. Second, our study is retrospective in design. A larger prospective study would be necessary to validate our findings. 

In conclusion, this study shows patients with DM-MDA5+ present an early infection over-risk and highlights the importance of monitoring for infection during patient follow-up. Further work is needed to determine the underlying pathophysiological mechanisms and to specify the role of the anti-MDA5 Abs in the disease. 

## Figures and Tables

**Figure 1 biomedicines-10-03176-f001:**
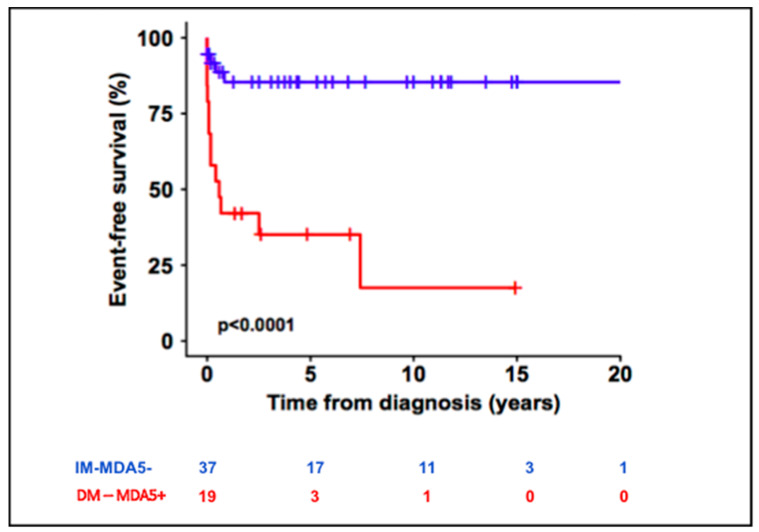
Significant infectious complications (SIC)-free survival curve in DM-MDA5+ and IM-MDA5− patients.

**Table 1 biomedicines-10-03176-t001:** Patient characteristics at baseline and underlying immunosuppression (regardless of DM/IM treatment).

	Whole Cohort (n = 56)	DM-MDA5 Positive (n = 19)	IM-MDA5 Negative (n = 37)	*p*-Value
**Demography**				
Male, n (%)	27 (48)	10 (53)	17 (46)	0.85
Age at diagnosis, years	46.2	52.5	42.9	0.2
**IM clinical presentation, n (%)**				
**General signs**				
Fever	10 (18)	6 (32)	4 (11)	0.12
Weight loss (>5% of the weight)	16 (29)	9 (47)	7 (19)	0.05
**Muscle**				
Muscle deficiency	18 (32)	6 (32)	12 (32)	1
Deglutition disorders	8 (14)	5 (26)	3 (8)	0.15
Dysphonia	3 (5)	2 (11)	1 (3)	0.55
CK elevation > 5000 UI/L	2 (4)	1 (6)	1 (3)	1
**Skin**				
Facial erythema	14 (26)	8 (44)	6 (17)	0.06
Digital ulcerations	8 (15)	8 (44)	0 (0)	0
**Lung involvement**				
CT abnormalities	31 (55)	11 (58)	20 (54)	1
Ground glass opacities	18/31 (33)	5/11 (26)	13/20 (36)	0.66
Pulmonary fibrosis	17/31 (31)	6/11(32)	11/20 (30)	1
Pneumomediastinum	2/31 (3)	1/11 (5)	1/20 (3)	1
**Functional assessment**				
Restrictive disorder	13 (37)	5 (39)	8 (36)	1
KCO (%)	41.3 ± 44	51.6 ± 39.9	36 ± 46.4	0.41
DLCO (%)	25.3 ± 26.5	37.1 ± 24.3	17.6 ± 25.8	0.08
Oxygen	7 (12.5)	4 (21)	3 (8)	0.34
Orotracheal intubation	4 (7)	1 (5)	3 (8)	1
**Heart**				
Pulmonary hypertension	1 (3)	1/16 (6)	0/15 (0)	1
**Biological parameters**				
Lymphocyte count (/mm^3^)	1342 ± 699	1106 ± 749.1	1472 ± 647.2	0.11
Neutrophils (mm^3^)	5679.2 ± 3428.9	4055 ± 3232.9	6607 ± 3234.6	0.02
IgG (g/L)	13.2 ± 2.5	13.4 ± 2.9	12.2 ± 0.9	0.30
**Comorbidities**				
Diabetes mellitus	5 (9)	1 (5)	4 (11)	0.85
Chronic kidney disease	1 (2)	1 (5)	0 (0)	0.73
Chronic hepatopathy	1 (2)	0 (0)	1 (3)	1
Chronic pulmonary pathology	1 (2)	1 (5)	0 (0)	0.73
Active smoking/stopped < 3 years	12 (21)	5 (26)	7 (19)	0.77
Cancer or Hematologic malignancy	9 (16)	4 (21)	5 (14)	0.73
**Cancer therapy**				
Surgery	6 (11)	3 (16)	3 (8)	0.67
Radiotherapy	5 (9)	3 (16)	2 (5)	0.43
Chemotherapy	6 (11)	3 (16)	3 (8)	0.67
**Others**				
HIV	0 (0)	0 (0)	0 (0)	1
Splenectomy	1 (2)	1 (5)	0 (0)	0.73
Iatrogenic neutropenia	2 (4)	1 (5)	1 (3)	1
Anterior immunosuppressive therapy	4 (7)	1 (5)	3 (8)	1
**Prophylaxis**				
Pneumocystis prophylaxis	19/54 (35)	12 (63)	8/35 (23)	0.01
Cotrimoxazole	17/54 (31)	11 (92)	7/35 (20)	0.01
Pentacarinat	2 (4)	1 (8)	1 (3)	0.62
Pneumococcal vaccination	21/51 (41)	9 (47)	12/32 (38)	0.69

CK: creatine kinase; ILD: interstitial lung disease; MDA5: melanoma differentiation-associated protein 5; RFTs: respiratory function tests; DLCO: diffusing capacity of the lung for carbon monoxide, KCO: diffusion coefficient. Chronic hepatopathy: related to chronic viral hepatitis (hepatitis B and C), alcohol or metabolic cirrhosis; paraneoplastic: cancer previous year or at the same time as the diagnosis of myositis.

**Table 2 biomedicines-10-03176-t002:** Clinical features of the SIC in both groups.

	Whole Cohort (n = 56)	Patients MDA 5 Positive (n = 19)	Patients MDA 5 Negative (n = 37)	*p*-Value
Follow-up time (months; mean ± sd)	70 ± 59	54 ± 45	79 ± 64	0.22
Number of SIC per patient	0.6 ± 1.14	1.4 ± 1.6	0.16 ± 0.4	<0.001
One year SIC-free infection, %	-	42.1 (+/− 11.3)	85.3 (+/− 6.1)	<0.001
Type of infection, n (%)				
*Microbial characteristics*				
Bacterial	12 (36)	9/27 (33)	3/6 (50)	0.44
Viral	5 (15)	3/27 (11)	2/6 (33)	0.46
Fungal	5 (15)	4/27 (15)	1/6 (17)	0.97
Pneumocystis	3 (9)	2 (7)	1 (17)	0.48
Undetermined infectious agent	11 (33)	11/27 (41)	0/6 (0)	0.055
*Site of infection*				
Lungs	16 (49)	13/27 (48)	3/6 (50)	0.93
Skin	8 (24)	6/27 (22)	2/6 (33)	0.57
Urinary tract	2 (6)	1/27 (4)	1/6 (17)	0.23
Joint	2 (6)	2/27 (7)	0/6 (0)	0.49
ENT infection	2 (6)	2/27 (7)	0/6 (0)	0.49
Septicemia	3 (9)	3/27 (11)	0/6 (0)	0.39
*Serious infection, n (%)*				
Intensive care	9 (27)	8 (30)	1 (17)	0.5

ENT: ear nose throat.

## Data Availability

Data is contained within the article or Appendix A.

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
