# Peer review of "Infection Risk in Patients with Dermatomyositis Associated with Anti-MDA5 Antibodies: A Historical Cohort Study"

_biomedicines, 2022, doi:10.3390/biomedicines10123176_

Round 1

Reviewer 1 Report

The only concern is the small sample size, but the manuscript is generally well written.

There are no concerns about the method of randomized comparison. The discussion reflects research results.

I think the readers will pay more attention to the events for infectious diseases in MDA-5 positive DM.

Author Response

General Comments

The manuscript has been revised to address all the reviewer comments. Specifically: (i) the misleading term " undocumented infection" has been changed in the revised manuscript by the term “infection by undetermined infectious agent”, as suggested by the reviewer 2; (ii) we have now expanded the discussion part of the hypotheses that may explain that DM-MDA5+ patients are more prone to infection, as suggested by the reviewer 2; (iii) statistical changes have been made, as suggested by the reviewer 2; (iii) we corrected typos. Revisions in the manuscript are marked in red font.

Referee’s comment: The only concern is the small sample size, but the manuscript is generally well written.There are no concerns about the method of randomized comparison. The discussion reflects research results. I think the readers will pay more attention to the events for infectious diseases in MDA-5 positive DM.

Our response: We thank the reviewer for his/her positive comment.

Reviewer 2 Report

The authors analyzed the prevalence of infections in patients with MDA5-positive dermatomyositis. The study is interesting as it provides additional data on the natural course of MDA5 patients. However, I have several remarks to the authors:

1. It is unrealistic that the following values are similar: CD4 T cell count (372.5 ± 357.1 / mm3 vs. 1079.3 ± 1219.2 129 /mm3). There is either a mistake in the calculation or a wrong test has been employed to analyze the difference. Even SD is three times higher in the second group than in the first.  Please double check. Otherwise, please provide an analysis of the normal distribution of analyzed values.

2. The term of " undocumented infection" is misleading. Was the infection documented in the medical charts or not? I would suggest changing the term to infection of an undetermined infectious agent or something similar. 

3. The authors provided some explanation as, to why patients with MDA5 antibodies are more prone to infections but maybe they can elaborate on it a little bit more. 

4. Please change the word "aspecific" to "a specific". Otherwise, please explained why aspecific assay was used.

5. In the authors discussion should address the limitation of the study (e.g. small sample size, retrospective nature of the study, etc.) 

Author Response

General Comments:

The manuscript has been revised to address all the reviewer comments. Specifically: (i) the misleading term " undocumented infection" has been changed in the revised manuscript by the term “infection by undetermined infectious agent”, as suggested by the reviewer 2; (ii) we have now expanded the discussion part of the hypotheses that may explain that DM-MDA5+ patients are more prone to infection, as suggested by the reviewer 2; (iii) statistical changes have been made, as suggested by the reviewer 2; (iii) we corrected typos. Revisions in the manuscript are marked in red font.

Referee’s comment: The authors analyzed the prevalence of infections in patients with MDA5-positive dermatomyositis. The study is interesting as it provides additional data on the natural course of MDA5 patients.

Our response: We thank the reviewer for his/her positive comment.

Point 1. It is unrealistic that the following values are similar: CD4 T cell count (372.5 ± 357.1 / mm3 vs. 1079.3 ± 1219.2 129 /mm3). There is either a mistake in the calculation or a wrong test has been employed to analyze the difference. Even SD is three times higher in the second group than in the first.  Please double check. Otherwise, please provide an analysis of the normal distribution of analyzed values.

Response 1: We thank the reviewer for bringing our attention to this point. We have been back to the raw data, and identified that the mistake was related to the amount of missing data for this parameter. We have removed this from the article.

Point 2. The term of " undocumented infection" is misleading. Was the infection documented in the medical charts or not? I would suggest changing the term to infection of an undetermined infectious agent or something similar. 

Response 2:  We thank the reviewer for his/her comment. We agree that the term "undocumented infection" was not appropriate,  as for all of our patients, infections were documented in the medical charts. The term "undocumented" was used in reference to the fact that no infectious agent was identified in the usual microbiological investigations despite the established infectious syndrome. As suggested by the reviewer, the term “undocumented infection” has been changed in the revised manuscript by the term “infection by undetermined infectious agent”.

Point 3. The authors provided some explanation as, to why patients with MDA5 antibodies are more prone to infections but maybe they can elaborate on it a little bit more. 

Response 3: We thank the reviewer for his/her comment. As the reviewer pointed out, we have already discussed of autoAbs as a possible cause that may explain why anti-MDA5+ patients are more prone to infection. We have now expanded this section by also discussing the fact that neutrophil count was lower in DM-MDA5+ patients. Line 260 of the revised manuscript: “An alternative hypothesis that could potentially explain that DM-MDA5+ patients are more susceptible to infections could be the lower number of neutrophils observed in these patients. This can be explained in particular by the dysregulation of neutrophils that takes place in the disease. Indeed, it has recently been shown that neutrophils from DM-MDA5+ patients are more prone to NETosis, a form of regulated cell death during which neutrophil extracellular traps (NET) and enzymes are extruded from the cell (21). This abnormal NETosis appears to be induced by autoAbs, although the precise pathophysiological mechanisms are still poorly understood (21). A dysregulation of this important process of the body's defense against infectious agents could thus contribute to facilitating infections in DM-MDA5+ patients.”

Point 4. Please change the word "aspecific" to "a specific".Otherwise, please explained why a specific assay was used.

Response 4: We apologize for this mistake and thank the reviewer for highlighting it. The correct term is: "a specific". The manuscript revised has been revised accordingly.

Point 5. In the authors discussion should address the limitation of the study (e.g. small sample size, retrospective nature of the study, etc.) 

Response 5: We thank the reviewer for his/her comment. Line 272 of the revised manuscript: "The current study had several limitations. First, the sample size was relatively small, which is explained by the fact that DM-MDA5+ is a rare disease, especially in Western countries. Second, our study is retrospective in design. A larger prospective study would be necessary to further investigate these findings".

Round 2

Reviewer 2 Report

None